# Graphite Nanoplatelets from Waste Chicken Feathers

**DOI:** 10.3390/ma13092109

**Published:** 2020-05-02

**Authors:** Bryan Pajarito, Amelia Jane Belarmino, Rizza Mae Calimbas, Jillian Rae Gonzales

**Affiliations:** Department of Chemical Engineering, University of the Philippines, Diliman, Quezon City 1101, Philippines; abbelarmino@up.edu.ph (A.J.B.); rdcalimbas@up.edu.ph (R.M.C.); raejill0910.edu@gmail.com (J.R.G.)

**Keywords:** graphite nanoplatelet, graphitization, carbon, chicken feather, exfoliation

## Abstract

Graphite nanoplatelets (GNPs), a functional 2D nanofiller for polymer nanocomposites, utilize natural graphite as a raw material due to its stacked graphene layers and outstanding material properties upon successful exfoliation into nano-thick sheets. However, the increasing demand for natural graphite in many industrial applications necessitates the use of graphite from waste resources. We synthesized GNPs from waste chicken feathers (WCFs) by graphitizing carbonized chicken feathers and exfoliating the graphitic carbon by high-speed homogenization and sonication. We then separated GNP from non-exfoliated carbon by centrifugation. This paper describes the morphology, chemical, and crystalline properties of WCF and its carbon derivatives, as well as the structural features of WCF-derived carbons. We obtained GNPs that have a 2D structure with huge variations in particle size and thickness. The GNP shows the presence of carbonyl groups, which are mostly attached at the edges of the stacked graphene sheets. Defects in the GNP are higher than in graphene synthesized from direct exfoliation of natural graphite but lower than in graphene oxide and reduced graphene oxide. To produce GNP of high quality from WCF, restacking of graphene sheets and concentration of carbonyls must be minimized.

## 1. Introduction

With their commercial availability, low cost, and structure similarity to graphene, graphite nanoplatelets (GNPs) continue to be a significant component of modern polymer nanocomposites [1,2,3,4,5,6]. These 2D nanofillers consist of stacked layers of graphene, with thickness ranging from a few to a hundred nanometers [1,2,4,5,6,7,8,9,10,11]. GNPs have high aspect ratio and surface area, low mass density, and excellent thermal, electrical, and mechanical properties [2,3,5,6,8,9,10,12,13,14]. Recent application of GNPs in different polymers demonstrates improvement in thermomechanical characteristics [7,15,16,17,18,19,20,21,22,23], electrical behavior [15,17,20,24], and corrosion resistance [25] of nanocomposites.

GNPs have been prepared previously, using different methods [5,8], but the exfoliation of intercalated graphite (GIC), followed by mechanical agitation, remains to be the most common approach [1,2,4,5,7,9,10,11,13,14,15,17,23,24,25,26,27,28,29,30]. In this method, GIC (natural graphite intercalated with acids or alkali metals) undergoes exfoliation by thermal treatment or microwave irradiation, to produce exfoliated graphite (EG). Mechanical agitation of EG in a solvent, usually by sonication, produces GNPs. At present, natural graphite serves as the main precursor of GNPs. However, the demand for natural graphite is continuously growing due to its many industrial applications, such as in refractories, lubricants, brakes, batteries, and chemically resistant materials [31,32]. As an important and strategic raw material, the widespread and heavy exploitation of natural graphite will result in its resource depletion [33]. The use of graphite derived from waste resources can alleviate the supply shortage and promote sustainability in the production of GNPs, graphene, and other related industries.

To contribute to this effort, we have prepared GNPs from carbonized and graphitized waste chicken feathers (WCFs). Recently, carbon-based nanomaterials have been produced from WCFs [34,35] and, in general, from waste biomass [36,37,38,39,40], to demonstrate the utilization of carbon feedstocks unrelated to fossil fuels. In the EU, WCFs and other keratinaceous wastes amount to 5 × 10^6^ t annually [41]. Worldwide, the poultry meat processing industry generates 40 × 10^6^ t of WCFs every year [42]. While there is a huge supply, the demand for WCFs is low and most are burned, landfilled, or converted into feather meal as livestock feed or fertilizer [43]. The mass accumulation of WCFs and its slow degradation in nature pose an environmental problem [44]. However, with keratin as the main component and with a carbon content of more than 40 wt% [35], WCF is a potential precursor of GNP. Carbonization of WCF produced carbonaceous biochar (carbonized chicken feather or CCF), which was graphitized at low temperature and acidic conditions, using an iron catalyst. The graphitized chicken feather (GCF) underwent liquid-phase exfoliation, to yield GNPs. We employed high-speed homogenization, sonication, and centrifugation to facilitate the exfoliation of GCF into GNPs [45,46]. This paper reports the morphology, chemical, and crystalline properties of WCF and its carbon derivatives (CCF, GCF, and GNP). This paper also presents the structural characteristics of WCF-derived carbons obtained by Raman spectroscopy.

## 2. Materials and Methods

WCFs were obtained from a poultry processing plant in Bulacan, Philippines. Hydrochloric acid (36 vol%, Univar, Ajax Finechem, Melbourne, Australia) and FeCl_3_·6H_2_O (AR grade, Techno Pharmchem, Delhi, India) were used in the graphitization of CCF. Moreover, 2-propanol (ACS Basic, Scharlau, Barcelona, Spain) and deionized water were utilized as a solvent during high-speed homogenization of GCF. 

WCFs were washed several times with water mixed with liquid detergent, to remove blood, manure, and other unwanted components. The washed WCFs were laid out on galvanized iron sheets and sun-dried for 3 days. WCFs were further conditioned in a laboratory oven (YLD 2000, Labtron Trading, Hampshire, UK) at 105 °C for 24 h before being cut into ~10 mm long pieces. After manually separating the rachis, the barbs were cut into ~2 mm long fibers and milled in an analytical mill (A11 Basic, IKA, Staufen, Germany) for 10 min. The resulting WCF powder was sieved (mesh 100) before carbonization.

The carbonization and graphitization of WCF were performed by following the method of Akhavan et al. [47]. Briefly, 5 g WCF was placed and covered in a crucible with limited air access. Covered crucibles containing WCF were transferred into a muffle furnace (FN 1208, Constance, London, UK) for carbonization at 400 °C for 5 h. The yield of CCF from WCF was 39.6 wt%. CCF was ground into a powder and sieved (mesh 100) before graphitization. CCF and FeCl_3_·6H_2_O (at 2:1 carbon-catalyst mass ratio) were added to distilled water (100 mL g^−1^ CCF). The pH of the solution was adjusted ~2 by adding HCl. The mixture was heated at 60 °C for 5 h, using a hot plate (MSH-30D, Daihan Scientific, Gangwon, South Korea), while being continuously homogenized at 1000 rpm by a high-speed homogenizer (T50 Digital Ultra-Turrax, IKA) equipped with a dispersing element (S50N-G45G, IKA). GCF was separated from the homogenized solution by vacuum filtration and dried in a laboratory oven at 105 °C for at least 48 h before further use.

To prepare GNP, 10 g GCF was added to 1000 mL of 40 wt% 2-propanol aqueous solution and was subjected to high-speed homogenization at 5000 rpm for 90 min. The homogenized solution was probe sonicated (Q700, QSonica, Newton, CT, USA) under an ice bath at amplitude 80 for 90 min. GNP was separated from non-exfoliated GCF by centrifugation (Rotofix 32A, Hettich, Tuttlingen, Germany) at 500 rpm for 120 min. The upper 30 mL of the centrifuged mixture was pipetted to several watch glasses and dried in a hot plate at 100 °C, to recover the brown GNP powder. The concentration of GNP in the pipetted solution was estimated to be 0.6 mg mL^−1^, as measured by drop-casting and drying until constant weight. The yield of GNP from GCF was 4.8 wt%.

The morphologies of WCF, CCF, and GCF were examined by using a scanning electron microscope (SEM; Zeiss UltraPlus) operated at 2 kV. The samples were mounted on a carbon tape placed on an aluminum stub. A small amount of GCF was diluted in ethanol, drop-casted, and dried in a sample stub before viewing in a field emission transmission electron microscope (TEM; JEOL JEM-2100F) operated at 200 kV. GNP solution was drop-casted and dried in a clean glass slide before examining in an atomic force microscope (AFM; XE-70 Park Systems, Santa Clara, CA, USA) at non-contact mode. Two-dimensional images were generated at 10 × 10 *μ*m scan size. Analysis of line profiles was performed by using XEI software (1.7.6, Park Systems, Santa Clara, CA, USA). The particle size distribution of GNPs was determined by using ImageJ software (1.8.0, National Institutes of Health, MD, USA). 

The chemical structures were studied, using a Fourier transform infrared spectrometer (FTIR; Frontier-IR, PerkinElmer, Waltham, MA, USA) coupled with a diamond/ZnSe attenuated total reflectance (ATR) crystal plate accessory. Four scans were accumulated for each measured spectrum, with a range of 650–4000 cm^−1^ and resolution of 4 cm^−1^. The background spectrum was obtained without the sample prior to the actual measurements.

The crystallinity of the samples was analyzed by X-ray diffraction (XRD; D2 Phaser, Bruker, Billerica, MA, USA), using Cu-K*α* radiation (*λ* = 0.154 nm) at 30 kV and 10 mA. Diffraction patterns were recorded from 2*θ* = 5°–80°, with an increment of 0.04°. Moreover, *d*-spacing and stacking height (*L_c_*) of graphitic carbon detected in GNP and GCF were determined by using Bragg and Scherrer equations [48]. The Bragg angle and the full width at half maximum (FWHM) were obtained by fitting a Lorentzian function to the diffraction peak, in the range of 2*θ* = 10–40°, using GRAMS/AI software. Impurities in CCF were determined by X-ray fluorescence (XRF; HD Mobile Analyzer, XOS Inc., East Greenbush, NY, USA) operated at 25–50 kV and 200 μA for an analysis area of 1 mm.

Raman spectra of carbon samples were taken on a micro-Raman spectrometer (Renishaw inVia, Gloucestershire, UK) with a 50 mW, 532 nm diode laser for excitation. Raman spectra were recorded from 800 to 2000 cm^−1^ and analyzed by using GRAMS/AI. After manual baseline correction, curve-fitting was performed by using the Levenberg–Marquardt algorithm. The spectra were deconvoluted, using Gaussian functions, and best-fitted with 5 bands.

## 3. Results and Discussion

The morphology of WCF (Figure 1a) shows a large amount of long and fibrous barbs attached to thick and stiff rachis, which is the main shaft of the feather. As natural protein fibers, barbs (Figure 1b) have a fibrillar surface and a honeycomb cellular cross-section [49]. Carbonization of WCF to form CCF melted the barbs and formed particulates of varied size and shape (Figure 2a). The volatilization of different components at 200–375 °C during carbonization [50] resulted in pores at the surface of CCF (Figure 2b). 

High-speed homogenization applied during graphitization of CCF to produce GCF decreased the particle size (Figure 3a). Moreover, the surface of GCF shows the growth of short and clustered micron-sized tubules (Figure 3b). Previously, in [51], nanotubes were developed on the surface of carbonized sawdust after iron-catalyzed graphitization by pyrolysis. The difference in the size of grown graphitic tubules could be attributed to the amount and type of catalyst used relative to carbon precursor, e.g., 0.002 mol FeCl_3_·6H_2_O g^−1^ CCF and 0.0001–0.001 mol Fe(NO_3_)_3_·9H_2_O g^−1^ sawdust in [51], and the type of graphitization method employed. Nevertheless, both studies demonstrate the formation of graphitic tubules due to iron catalyst particles etching through the bulk carbon. Graphitization has also only occurred at the surface of CCF. In the nanoscale, GCF shows the aggregation of 2D nanoplatelets (Figure 4a). The thickness of these nanoplatelets varies within a cluster of GCF (Figure 4b). 

The liquid-phase exfoliation of GCF to yield GNP resulted in 2D nanomaterials of varied particle sizes and thicknesses (Figure 5). Most of the GNPs have a particle size of less than 1 um (Figure 6). During high-speed homogenization, GCF particles were drawn axially into the dispersion head and ejected radially through the slots of the rotor-stator assembly [52]. The strong shear and thrust forces experienced by GCF facilitated its size reduction and extensive exfoliation. Sonication further assisted the exfoliation process, and centrifugation effectively separates the suspended GNP from non-exfoliated GCF, which settled at the bottom of the container. Still, we observed a huge variation in the thickness of the GNPs, with most of the particles remaining thick after exfoliation. This is due to the absence of intercalant and the high tendency of the graphene sheets to restack [53].

The chemical structure of GNP (Figure 7) indicates the stretching vibrations of C=C and C=O of carboxylic acids and carbonyl groups (strong intensity at 1630 cm^−1^) [54,55]. The brownish color of GNP observed after exfoliation of black GCF indicates graphite oxidation [54]. The carbonyl, which is also prominent in GCF and CCF, originates from the peptide bonds (–CONH of Amide I) of WCF [56,57]. Carbonizing WCF to produce CCF resulted in the scission of Amide A (3280 cm^−1^) and Amide II (1533 cm^−1^) [58] in the keratin structure. Retention of the carbonyl group in CCF suggests incomplete carbonization of WCF. Meanwhile, residual moisture remained in GCF (broad peak at 3273 cm^−1^), which was eliminated after its exfoliation and drying to produce GNP.

Both GNP and GCF show a graphite-like arrangement (002 diffraction peak around 2*θ* = 25°) in their crystalline structures (Figure 8). The lowered and broadened diffraction intensity of GNP relative to GCF indicates a high degree of exfoliation [59]. GNP has a smaller *d*-spacing than GCF (0.348 nm for GNP; 0.355 nm for GCF), resulting from the mild evaporation of intercalated H_2_O molecules during thermal annealing of GNP at 100 °C [60]. Moreover, GNP has slightly higher *Lc* than GCF (2.1 nm for GNP; 1.6 nm for GCF) as a result of restacked graphene sheets due to thermal annealing [53,61,62]. These XRD results are consistent with the observed C=C bonds from the ATR–FTIR findings (Figure 7).

CCF also shows graphitic ordering, as observed previously for pyrolyzed feathers [58]. Other diffraction peaks of CCF correspond to different calcium compounds, e.g., CaO (32.6°), Ca(OH)_2_ (34.5°), and CaCO_3_ (24.2°, 27.0°, and 32.6°) [63,64]. Results of XRF confirm the presence of calcium in CCF (Cl-5969, Ca-3212, Zn-290, Fe-140, Cu-17.6, and Sr-14.5 ppm). CaCl_2_ and Na_2_CO_3_ in detergent [65] have remained in the WCF after washing, which led to the formation of the said compounds during carbonization. Note that the calcium impurities are no longer present in GNP and GCF, as graphitization and exfoliation (both performed in aqueous solution) removed these impurities. Besides the formation of graphitic carbon and calcium compounds, carbonization also caused the disappearance of *α*-helix (10.2°) and *β*-sheet (20.7°) crystal structures of peptide chains in WCF [56].

The Raman spectra of WCF-derived carbons (Figure 9) exhibit the following peaks: the G band (1560 cm^−1^) for the in-plane bond stretching motion of sp^2^ carbon; the D1 band (1339 cm^−1^) for the in-plane breathing vibration of sp^2^ carbon within structural defects; the D3 band (1433 cm^−1^) for the amorphous sp^2^ carbon; and the D4 band (1224 cm^−1^) for the sp^2^-sp^3^ bonds or C-C and C=C stretching vibration in polyenes [66,67,68,69,70]. The band B (1140 cm^−1^) denotes an artifact of the instrumental setup and is not related to the carbon samples [66]. Table 1 summarizes the results of peak deconvolution, including the position of the bands, FWHM, and the ratio of the band to the total Raman peak area (*I*/*I*_Total_). In all the samples, the G and D1 bands have the largest band areas as compared to the other bands. The FWHMs of G (154.0–167.7 cm^−1^) and D1 (128–140.4 cm^−1^) bands are larger than in natural and synthetic graphites (16–19 cm^−1^ for the G band; 38–52 cm^−1^ for the D1 band) but comparable to graphene oxide (114–133 cm^−1^ for the G band; 214–226 cm^−1^ for the D1 band) and thermally reduced graphene oxide (109–125 cm^−1^ for the G band; 198–217 cm^−1^ for the D1 band) [71]. This confirms the presence of oxygen functionalities in the carbon samples primarily as carbonyl groups (Figure 7). Considering the absence of a diffraction peak at 2*θ* = 11° for graphene oxide (Figure 8), the carbonyl groups must be present at the edges of the graphitic carbon and not at its interlayer space [60]. The wide G band of our samples indicates a high concentration of basal-plane defects [72]. The wide D1 band, which describes both basal-plane and edge defects [72], suggests the presence of heteroatoms or point dislocations in the structure [71].

All carbon samples show a well-ordered graphitic structure (*I*_G_/*I*_Total_ > *I*_D1_/*I*_Total_). GNP has the lowest *I*_D1_/*I*_G_ ratio, i.e., GCF (0.75) > CCF (0.71) > GNP (0.56), meaning that it has the lowest number of defects [71,72]. Although defects were introduced to GCF during homogenization and sonication to produce GNP, centrifugation separates GNP from large and unstably dispersed graphite particles and other aggregates [73]. Due to exfoliation, the centrifuged GNP is relatively thin compared to GCF, and CCF and has a lower quantity of edges of graphene sheets, as shown by a narrower FWHM of D1 band and lower *I*_D1_/*I*_Total_. The defect content of GNP is higher than most graphene reported in the literature, e.g., *I*_D1_/*I*_G_ < 0.18 [72,73] and 0.2–0.4 in [74], but lower than graphene oxide and reduced graphene oxide, where *I*_D1_/*I*_G_ > 1 [71,73]. Meanwhile, the high *I*_D1_/*I*_G_ of GCF, as well as the narrower FWHM of the G band and higher *I*_G_/*I*_Total_, compared to CCF, can be attributed to the defects brought by the growth of graphitic tubules on the surface of CCF.

## 4. Conclusions

GNP can be produced from WCF by the following procedure: (1) carbonize WCF to produce CCF; (2) graphitize CCF at low temperature and acidic conditions, using an iron catalyst to generate GCF; (3) homogenize and sonicate GCF, to exfoliate the graphitic carbon; and (4) separate the exfoliated graphite from non-exfoliated carbon by centrifugation to yield GNP. The obtained GNPs are 2D nanomaterials of varied particle size and thickness. The GNP has carbonyl groups attached at the edges of the stacked graphene sheets. Defects in GNP are higher than in graphene prepared from direct exfoliation of natural graphite but lower than in graphene oxide and reduced graphene oxide. To consider WCF as a sustainable source of graphite for high-quality GNP, further research is needed to lessen restacking of graphene sheets, lower the concentration of carbonyl groups, and reduce the defects in the GNP.

## Figures and Tables

**Figure 1 materials-13-02109-f001:**
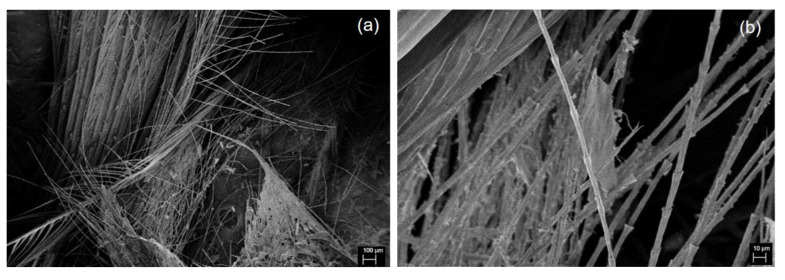
SEM images of WCF at (**a**) 100× and (**b**) 1000× magnification.

**Figure 2 materials-13-02109-f002:**
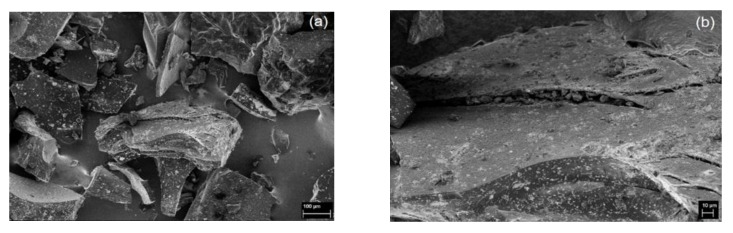
SEM images of CCF at (**a**) 250× and (**b**) 1000× magnification.

**Figure 3 materials-13-02109-f003:**
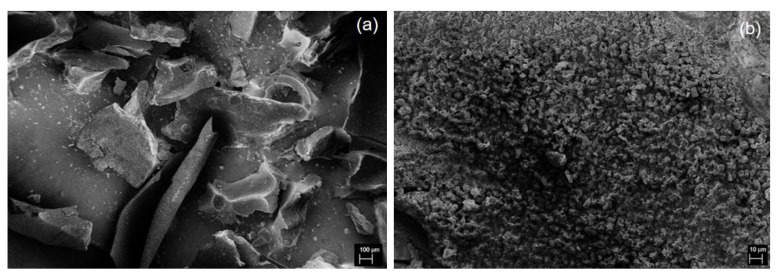
SEM images of GCF at (**a**) 100× and (**b**) 1000× magnification.

**Figure 4 materials-13-02109-f004:**
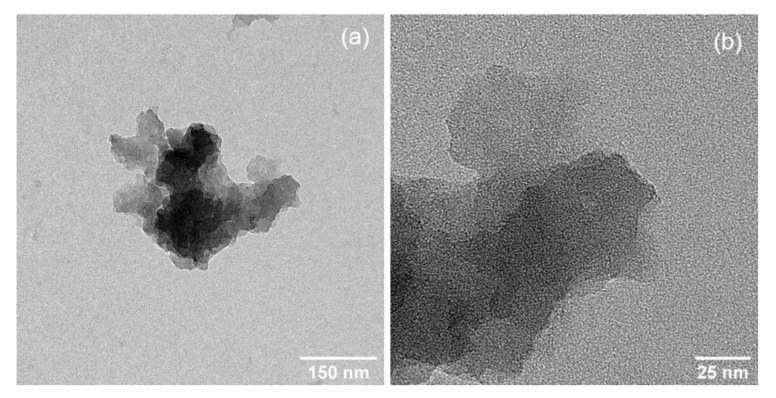
TEM images showing (**a**) a cluster of GCF composed of (**b**) aggregated 2D nanoplatelets.

**Figure 5 materials-13-02109-f005:**
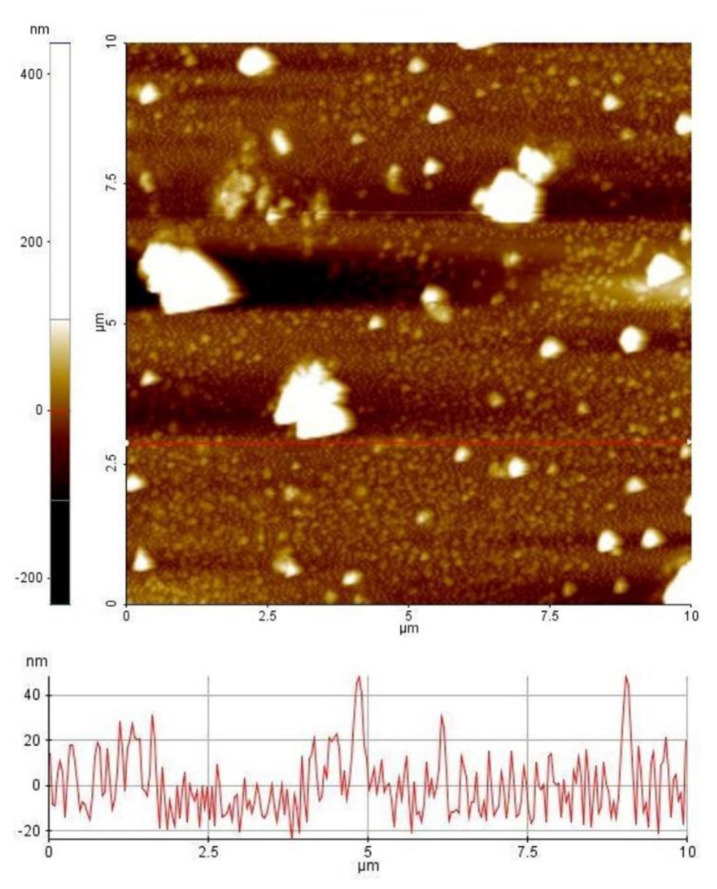
AFM image and line profile of GNPs.

**Figure 6 materials-13-02109-f006:**
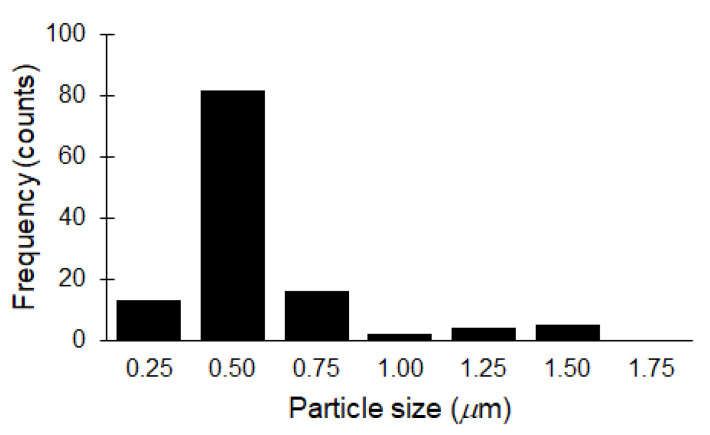
Particle size distribution of GNPs.

**Figure 7 materials-13-02109-f007:**
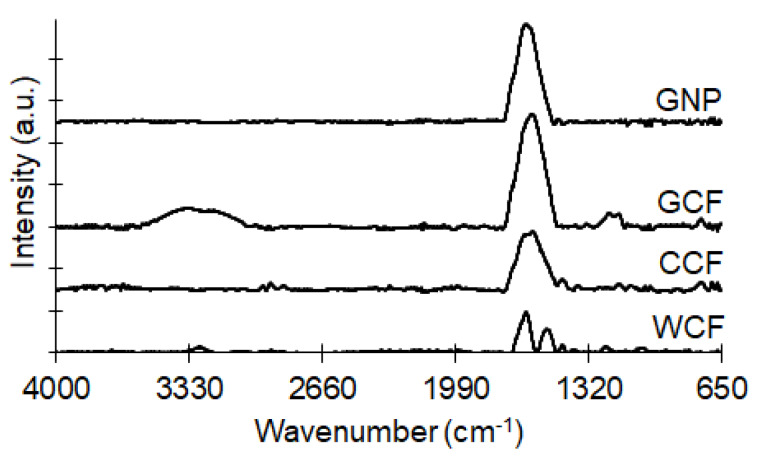
ATR–FTIR spectra of GNP, GCF, CCF, and WCF.

**Figure 8 materials-13-02109-f008:**
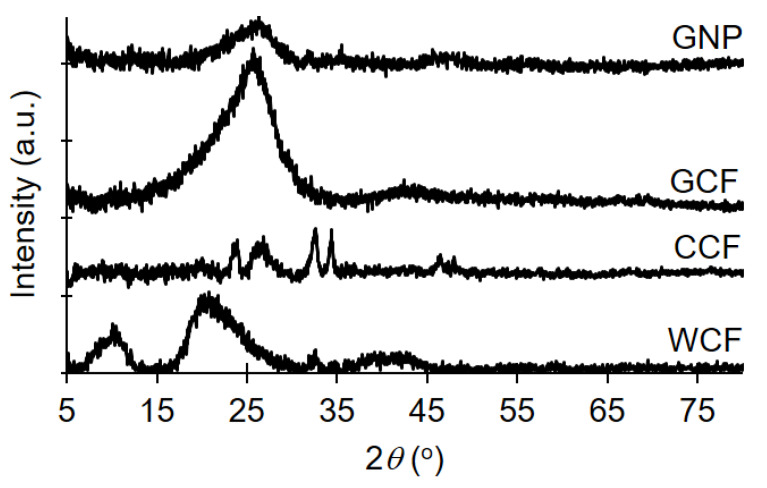
XRD patterns of GNP, GCF, CCF, and WCF.

**Figure 9 materials-13-02109-f009:**
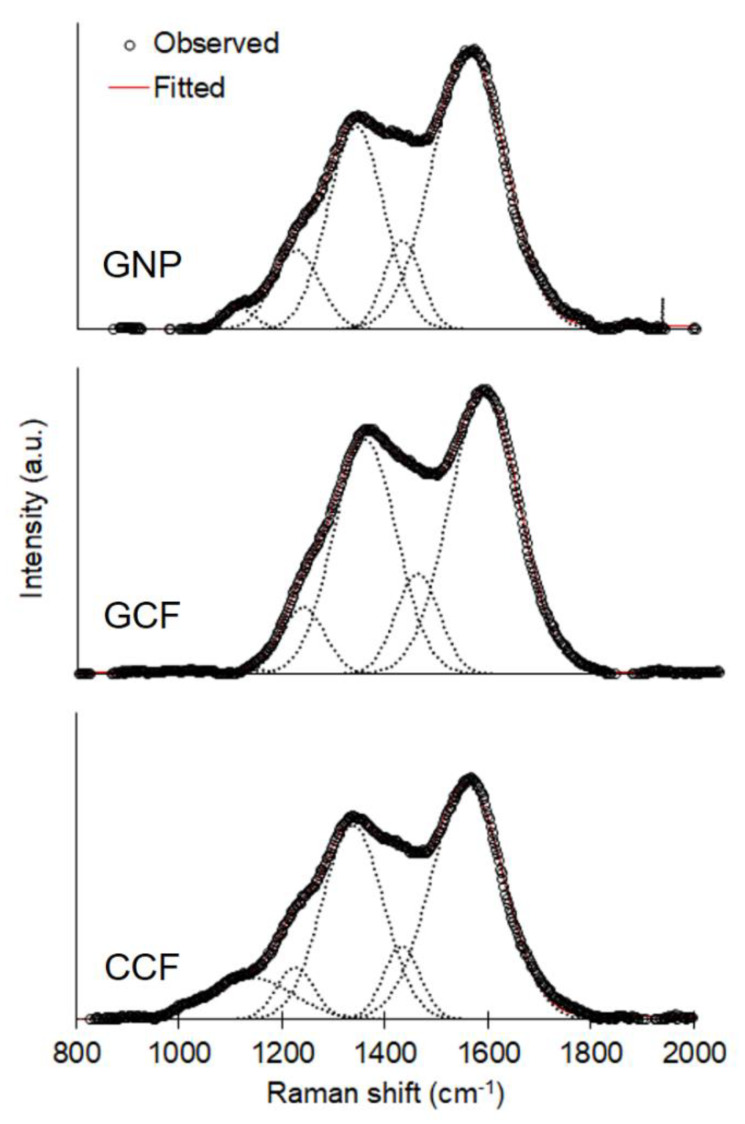
Raman spectra of GNP, GCF, and CCF.

**Table 1 materials-13-02109-t001:** Position, FWHM, and *I*/*I*_Total_ of peaks after deconvolution of Raman spectra.

Sample	Peak (cm^−1^)	FWHM (cm^−1^)	*I*/*I*_Total_ (%)
GNP	1560.1	167.7	49.7
	1431.1	84.3	8.1
	1340.6	128.0	27.8
	1224.4	108.1	9.2
	1114.0	71.6	1.6
GCF	1560.3	154.0	46.2
	1435.8	96.5	10.4
	1337.6	138.6	34.8
	1222.0	94.9	6.8
	1137.6	38.6	0.2
CCF	1560.1	165.1	44.2
	1432.9	84.4	7.1
	1337.8	140.4	31.2
	1225.5	86.7	5.2
	1140.1	191.6	9.2

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
