# Peer review of "Graphite Nanoplatelets from Waste Chicken Feathers"

_materials, 2020, doi:10.3390/ma13092109_

Round 1

Reviewer 1 Report

The manuscript reported synthesized graphite nanoplatelets from waste chicken feathers by graphitizing carbonized chicken feathers and exfoliating the graphitic carbon by high-speed homogenization and sonication. However, the manuscript lacks the novelty and innovativeness. Carbonizing chicken feathers causes serious environmental pollution and creates unpleasant odor, leading to the unsuitable application at a large scale. After careful evaluation, this manuscript is not recommended to be published in Materials.

  1. The fabrication process is complex, cumbersome, time-consuming, and difficult to scale up for practical use. Thus, graphite nanoplatelets from waste chicken feathers do not possess a commercial and ecological value.
  2. The icon is not clear and the format is not uniform, such as Figure 3 and Figure 4.
  3. The AFM analysis is not accurate in Figure 5, and the particles of graphite nanoplatelets are uneven, so the conclusion is not convincing.
  4. Raman spectra of GNP, GCF, and CCF in Figure 8 were almost identical, the ratio of IG/ITotal and ID1/ITotal are not given in this paper, and I/ITotal in Table 1 does not make much sense.

Reviewer 2 Report

This paper reports on preparation of Graphite Nanoplatelets from (Carbonized) Waste Chicken Feathers. From the viewpoint of recycle of resources as well as preparation method of nano-materials, it should be published in Materials essentially. However, improvement should be done before acceptance. (1) Effect of original morphorogy According to SEM images, morphorogy of reactants may be determine that of products. Please mention the effects of initial morphorogy of reactants. (2) Addition of carbon According to XRD patterns, CCF did not keep crystalline feature, it may affect not only addition of materials but also states of the material (WCF). Please mention the effect of treatment of addition of carbon. (3) Effect of impurity WCF is made of several elements except for carbon. How did they convert after formation of GNP? Please mention the effects of initial composition of reactants. That's all.

Reviewer 3 Report

The manuscript “Graphite Nanoplatelets from Waste Chicken Feathers” reports on the production of graphene-related materials by graphitizing waste chicken feathers. The idea is interesting. Unless the characterization was performed with the right methods, some of the results need improvements. Moreover, the introduction misses some important references. Major revisions are needed.

Introduction:

  • The authors should introduce and discuss more accurately the recent works on the production of carbon-based nanomaterials from chicken feathers and in general from wastes. See for example https://doi.org/10.1016/j.rser.2015.12.120,  https://doi.org/10.1002/jctb.5196.
  • The authors should introduce the concept of liquid-phase exfoliation and centrifugation since they use both. The authors should use appropriate references.
  • The authors could introduce better and with more numbers the problem of the keratinaceous wastes generated by industries (chicken but also other farm wastes). This will permit the reader to understand that their research is contributing to solving a real problem and will make the article more interesting. See for example https://doi.org/10.1021/acssuschemeng.9b02415.

Results

  • Can the authors provide a graph with the lateral size distribution of the obtained micro-nano platelets?
  • Figure 5: With such high thickness AFM is not the correct technique to measure the thickness.
  • Figure 7: Why the authors speak about a high degree of exfoliation?
  • Figure 8: The Raman spectra should be collected until 3000 cm-1 to see the 2D peaks. 
  • Why after sonication and ultracentrifugation the thickness of the nanoparticles is so high? Did the authors try to use different solvents, power of the sonicator and do multiple sonication-ultracentrifugation steps? The quality of their material is quite poor.
  • Is the obtained nanopowder electrically conductive? 

Reviewer 4 Report

The characterization methods for testing carbon materials and the starting material (chicken feathers) have been correctly selected and allow to track structural changes after subsequent stages of synthesis. The authors have drawn the right conclusions from analysis: X-ray diffraction spectra (XRD), Raman spectroscopy and morphology analysis using scanning electron microscopy (SEM), transmission electron microscopy (TEM) and force electron microscopy (AFM). One can have doubts about the interpretation of infrared spectroscopy analysis. The spectrum is not very accurate and the closely spaced peaks overlap to form a wide band covering a large range of vibrations. The spectrum in Figure 6 in the range 1500-1900 cm-1 contains closely placed peaks specific for carbonyl groups (C = 0) 1730 cm-1 and aromatic double bonds (C = C) 1630 cm-1, which are probably more intense than carbonyl groups. The appearance of C = C bonds is a evidence of the formation of graphene aromatic domains and is consistent with the XRD spectrum.

Round 2

Reviewer 1 Report

After cautiously review the responding letter, this paper can be considered to be accepted for publication.

Author Response

Thank you for reviewing our manuscript. 

Reviewer 3 Report

The authors substantially improved the paper.

The reviewer strongly suggest to investigate the 2D Raman peak.

Author Response

The authors substantially improved the paper.

Point: The reviewer strongly suggest to investigate the 2D Raman peak.

Response: As we have mentioned in our first response, the Raman data was provided by our collaborator (see Acknowledgments). Unfortunately, we no longer have access to the Raman equipment and thus, we can only present the first-order Raman characteristics of our carbon samples in the manuscript.

We would also like to add that the 2D Raman peak (which is around 2700 cm-1) is mainly used to confirm the number of graphitic layers in single and few-layered graphene (specifically 1-4 layers; see J Phys Chem C 119: 10653-10661 and also Phys Rep 473: 51-87). An isolated graphene sheet will give a single peak, while 2-4 layers in a Bernal structure will give multiple bands. Thick graphitic crystals will give a double band at D2. While we don't have the second-order Raman data, our SEM results for CCF and GCF already show that we have thick graphitic samples. And while not accurate, our AFM results also confirm the presence of thick graphitic samples in GNP. 

We would like to thank the reviewer for reviewing our manuscript.